:ۈ: PLOS | ONE

# Evaluation of four commercial tests for detecting ceftiofur in waste milk bulk tank samples

Marlene Belmar[1☉], Sharif Aly[2☉], Betsy M. Karle[3], Richard V. Pereira[1☉¤]*

**1** Department of Population Health and Reproduction, School of Veterinary Medicine, University of California Davis, Davis, CA, United States of America, **2** Veterinary Medicine Teaching and Research Center, School of Veterinary Medicine, University of California Davis, Tulare, CA, United States of America, **3** Cooperative Extension, Division of Agriculture and Natural Resources, University of California, Orland, CA, United States of America

☉ These authors contributed equally to this work.
¤ Current address: University of California Davis One Shield, Davis, CA, United States of America
* rvpereira@ucdavis.edu

**Data Availability Statement:** All relevant data are within the manuscript and its Supporting Information files.

**Funding:** Funding is provided by U.S. Department of Agriculture Project number CA-V-PHR-4708-

## Abstract

The objective of this study was to identify factors affecting the accuracy of four commercial tests for ceftiofur drug residue in milk samples from bulk tank waste milk (WM). WM samples were collected from 12 California dairy farms which were initially tested using liquid chromatography (LC-MS/MS) to confirm their negative status for drug residues above the FDA established tolerance/safe levels. The milk samples were also tested for fat, protein, lactose, solids non-fat (SNF), somatic cell count (SCC), coliform count, and standard plate count (SPC). Each WM sample was divided into two aliquots, one labeled as negative for drug residues (WMN) and the second spiked with ceftiofur as positive for ceftiofur residues (WMPos). Both types of WM samples were tested to evaluate the performance of 4 commercially available tests: Penzyme® Milk Test, SNAP® β–lactam, BetaStar® Plus and Delvo SP-NT®. Three assays in triplicates for the WMN and WMPos were conducted for each WM sample. Test were evaluated using sensitivity, specificity, positive predictive value, negative predictive value and positive likelihood ratio. Kruskal-Wallis method was used to evaluate the effect of milk quality parameters on true positive (TP) and false negative (FN) test results. All WMPos samples were identified as positive by all four tests, rendering 100% sensitivity for each test. The specificity for Penzyme, BetaStar, Delvo, and SNAP tests were 59.2, 55.5, 44.4, and 29.6, respectively. Overall, all tests correctly identified samples with ceftiofur residues (WMPos), as shown by 100% sensitivity. Greater variability was observed regarding identification of samples free of any drug residue, with Penzyme and BetaStar having the highest risk for correctly identifying TN samples. Our findings indicate that when selecting commercial tests to detect drug residues in WM, milk quality parameters must be considered if the aim is to reduce FP test results.

AH408 and University of California, Davis, Division of Agriculture and Natural Resources (Grant #1753). Any opinions, findings, conclusions, or recommendations expressed in this publication are those of the author(s) and do not necessarily reflect the view of the U.S. Department of Agriculture. The funder had no role in study design, data collection and analysis, decision to publish, or preparation of the manuscript.

**Competing interests:** The authors have declared that no competing interests exist.

## Introduction

Antimicrobial resistance is a great concern for human and animal health. Ceftiofur is a third-generation cephalosporin, a drug class of critical importance to both human and veterinary medicine; hence extending bacterial susceptibility to ceftiofur is a priority [1, 2]. Ceftiofur's broad antimicrobial spectrum explains its high demand in veterinary medicine. Moreover, ceftiofur is available in food animal formulations with no milk withdrawal periods, making it a preferred antimicrobial drug choice to treat dairy cattle infections.

Transition milk from recently calved dairy cows and milk from hospital cows that contains drug residues during treatment regimens are not fit for human consumption. Such milk, commonly identified as waste milk (WM), is harvested separately from saleable milk and utilized as a feed source for newborn calves by one-third of US dairy farms [3] and by 75.2% of CA dairies in a recent survey [4]. Two recent studies identified ceftiofur as the most common drug residue in WM samples from dairy farms in NY and CA [5, 6]. Furthermore, feeding calves WM containing ceftiofur residues at the concentrations observed on dairies has been shown to result in selection of *E. coli* resistant to ceftiofur and ceftriaxone, as well as other medically important drugs [7].

Current methods available for detecting drug residues in milk include microbial growth inhibition assays, microbial receptor assays, receptor binding assays, immunologic assays, enzymatic assays, and chromatographic analysis [8]. Such commercial tests are intended for use in raw, commingled saleable milk [9]. Although prior studies have evaluated the impact that many factors can have on test ability to correctly identify samples with drug residues (e.g. milk composition, SCC and total bacteria count), no study has evaluated these tests using bulk tank WM [10, 11]. The objective of this study was to identify factors affecting the accuracy of four commercial tests for ceftiofur drug residue in milk samples from bulk tank waste milk (WM).

## Material and methods

### Sample selection and processing

Waste milk samples were collected from 12 different commercial farms across California between September 2016 and March 2017. Waste milk samples were collected after verbal consent was obtained. Sample collection details are described previously [6]. Samples were initially tested using liquid chromatography–mass spectrometry/mass spectrometry (LC-MS/MS) to evaluate if they were negative for drug residues present above the FDA established tolerance/safe levels. The limits of quantification (**LOQ**) for the 27 drug residues screened using liquid chromatography–mass spectrometry/mass spectrometry (LC-MS/MS) are reported in **S1 Table**. Liquid chromatography is the gold standard for identifying and quantifying antimicrobial drug residues in milk samples (Gibbons-Burgener, et al., 2001). A pitfall of the LC-MS/MS screening panel used in this study is the lack of screening for amoxicillin, one of the drugs that could result in a positive test using the commercial tests being evaluated in the current study. The BetaStar (BetaStar® Plus, Neogen Animal Safety, Lexington, KY) test was used to identify samples that could be positive for amoxicillin. BetaStar was selected for ampicillin screening because of its ability to differentiate between ceftiofur and other beta lactams (amoxicillin, ampicillin, cephapirin, cloxacillin, and penicillin). Twelve milk samples collected in the original study were selected based on negative test results for beta-lactam drug residues in the milk using LC-MS/MS results. From these 12 samples, three were identified as being positive for amoxicillin drug residues using the BetaStar test, and were excluded from the study. Samples size calculation for detecting a difference between two means between false positive and true

negative samples for all milk quality parameters was conducted using this function in JMP (SAS Institute Inc., Cary, NC). The parameters and values used for this calculation were based on our current dataset, and output of this analysis is displayed in S2 Table.

Milk samples were also tested for fat, protein, lactose, solids non-fat (SNF) percent, somatic cell count (SCC, cells/ml), coliform count (CFU/ml), and standard plate count (SPC, CFU/ml) by the Sierra Dairy Laboratory (Tulare, CA). Each WM sample was divided into two aliquots, one labeled as negative for drug residues (WMN) and one as positive for ceftiofur residues (WMPos). The WMPos samples were WMN samples that were spiked with ceftiofur to a final concentration of 100 ppb of ceftiofur, which is the FDA tolerance for ceftiofur in saleable milk and should result in a positive test result using a commercial test for ceftiofur in saleable milk [12]. The samples (9 WMN and 9 WMPos) were tested to evaluate the performance of 4 commercially available tests: Penzyme® milk test (Neogen Animal Safety, Lexington, KY), SNAP β–lactam® (IDEXX Laboratories, Westbrook, ME), BetaStar Plus® (Neogen Animal Safety, Lexington, KY), and Delvo SP-NT® (DSM Food Specialties, South Bend, IN). These commercial test were not developed for screening of drug residues in waste milk samples, and therefore findings of this study should not be extrapolated to conditions to which these test were developed.

Negative and positive controls for ceftiofur residues were tested using pasteurized whole milk. Four assays in triplicate for the WMN and WMPos were conducted for each WM sample. All samples were vortexed for 30 seconds prior to testing. Samples were tested using the 4 commercial assays, following manufacturer's recommendations. Test results were labeled as true positive (TP) when resulting in a positive test result in WMPos samples, and false negative (FN) when a negative test result was obtained from WMPos samples. Test results were labeled as true negative (TN) when resulting in a negative test result in WMN samples. Samples were labeled as false positive (FP) when a positive test result was observed for WMN samples, indicating that although drug residues for β-lactam drug were below the FDA tolerance or safe levels for milk, the test still indicated that the sample was positive. In addition to the milk characteristics evaluated in the study that could affect test results, a drug residues detection limit below that established by the FDA could also potentially result in FP result. Milk drug residues screening test detection limits for Beta-lactam drugs used the study are shown in Table 1.

The assays were tested in triplicates for the WMN and WMPos for each WM sample for each of the four commercial tests, resulting in a total of 216 tests being conducted. Milk samples were thawed the night before analysis in a 4 °C refrigerator. All samples were vortexed for 30 seconds prior to testing.

## Statistical analysis

For each assay, sensitivity, specificity, positive predictive value, negative predictive value and positive likelihood ratio were calculated using standard methods [18]. For the dataset of the WMN samples that were not spiked, the effect of milk quality parameters on the within and between test variation of results indicating false-positive and true-negative test results were evaluated.

For each commercial test, analysis of variance models were created with the following dependent variables: fat percent, protein percent, lactose percent, SNF percent, SCC (cells/ml), coliform count (CFU/ml), and SPC (CFU/ml). The binary categorical variable for a test classified as false-positive or not, or true-negative or not was used as the explanatory variable. Assay number (repetitive measure) was controlled in the models as a random effect. Homoscedasticity was evaluated by plotting the standardized residuals versus predicted values, and supported

**Table 1. Milk drug residues screening test detection limits for Beta-lactam drugs.** Antimicrobial drug concentration at which greater than 90% of the replicates tested elicit a positive test response. Value for drugs are in parts per billion (ppb).

| TESTS | Drug Concentration (in ppb) [1] | | | | | | References |
|---|---|---|---|---|---|---|---|
| | Penicillin G | Ampicillin | Amoxicillin | Cloxacillin | Ceftiofur | Cephapirin | |
| Penzyme Milk Test | 5 | 7 | 6 | ND[2] | 80 | 11.6 | [13] |
| BetaStar Plus | 4.7 | 5.2 | 5.5 | 8.2 | 80 | 19 | [14] |
| SNAP β–lactam | 3 | 5.8 | 7.3 | ND[2] | 12 | 11.7 | [15] |
| Delvo SP-NT | 2 | 4 | 4 | 20 | 100 | 10 | [16] |
| FDA Tolerance/Safe Levels | 5 | 10 | 10 | 10 | 100 | 20 | [17] |

1. Antimicrobial drug concentration at which greater than 90% of the replicates tested elicit a positive test response.

2. ND, drug not screened by the test referenced.

the assumption for a linear regression model when points were reasonably equally distributed across all values of the independent variables. Normality assumption was evaluated by visual inspection of the data, distribution frequency (histogram), normal quantile plots and also by using the Shapiro-Wilk test. The equal variance assumption was evaluated by Levene and Welch's tests. When any of these assumptions were violated, the non-parametric Kruskal-Wallis test was used. A variable was considered significant when a $P$ value $\leq 0.05$ was observed. All analyses were conducted using JMP (SAS Institute Inc., Cary, NC).

## Results

Mean value (95% CI) for WM quality parameters (composition and microbiology) for samples used in the study (n = 9) were: 4.7% (3.4–5.9) fat, 3.7% (3.4–3.9) protein, 4.3% (4.1–4.5) lactose, 8.7% (8.5–8.9) SNF, 2,378 $10^4$ cells/ml (1,457–3,299) SCC, 293 CFU/ml (81–667) coliform count, and 92,000 CFU/ml (39,382–144,617) SPC (S3 Table).

All WMPos samples were identified as positive by all four tests, rendering 100% sensitivity. Results for specificity (Sp), positive predictive value (PPV) and positive likelihood ratio (LR+) are presented in Table 2. Least square means (LSM) results from linear regression for milk quality parameters by false positive (FP) and true negative (TN) status are depicted in Fig 1. All milk quality parameters rejected the test of normality, and the Kruskal-Wallis nonparametric approach was used. Results for this analysis found to significant difference between FP and

**Table 2. Overall specificity (Sp), positive predictive value (PPV), and positive likelihood ratio (LR+) for detection of ceftiofur residues in milk using four commercial assays.** All four tests had 100% sensitivity.

| | % Sp (95% CI)[1] | % PPV (95% CI)[1] | LR+ (95% CI)[1] |
|---|---|---|---|
| Penzyme[2] | 59.2 (43.3–75.2) | 71.1 (62.8–79.4) | 2.5 (1.4–3.5) |
| BetaStar[3] | 55.5 (N/A)* | 69.2 (N/A)* | 2.2 (N/A)* |
| Delvo[4] | 44.4 (N/A)* | 64.3 (N/A)* | 1.8 (N/A)* |
| SNAP[5] | 29.6 (13.7–45.5) | 58.7 (53.3–64.1) | (1.1–1.7) |

1. 95% confidence interval for results from triplicate testing.

2. Penzyme® Milk Test.

3. BetaStar® Plus.

4. Delvo®- SP.

5. SNAP® β–lactam.

* no variability in triplicate testing results (complete agreement between triplicate testing).

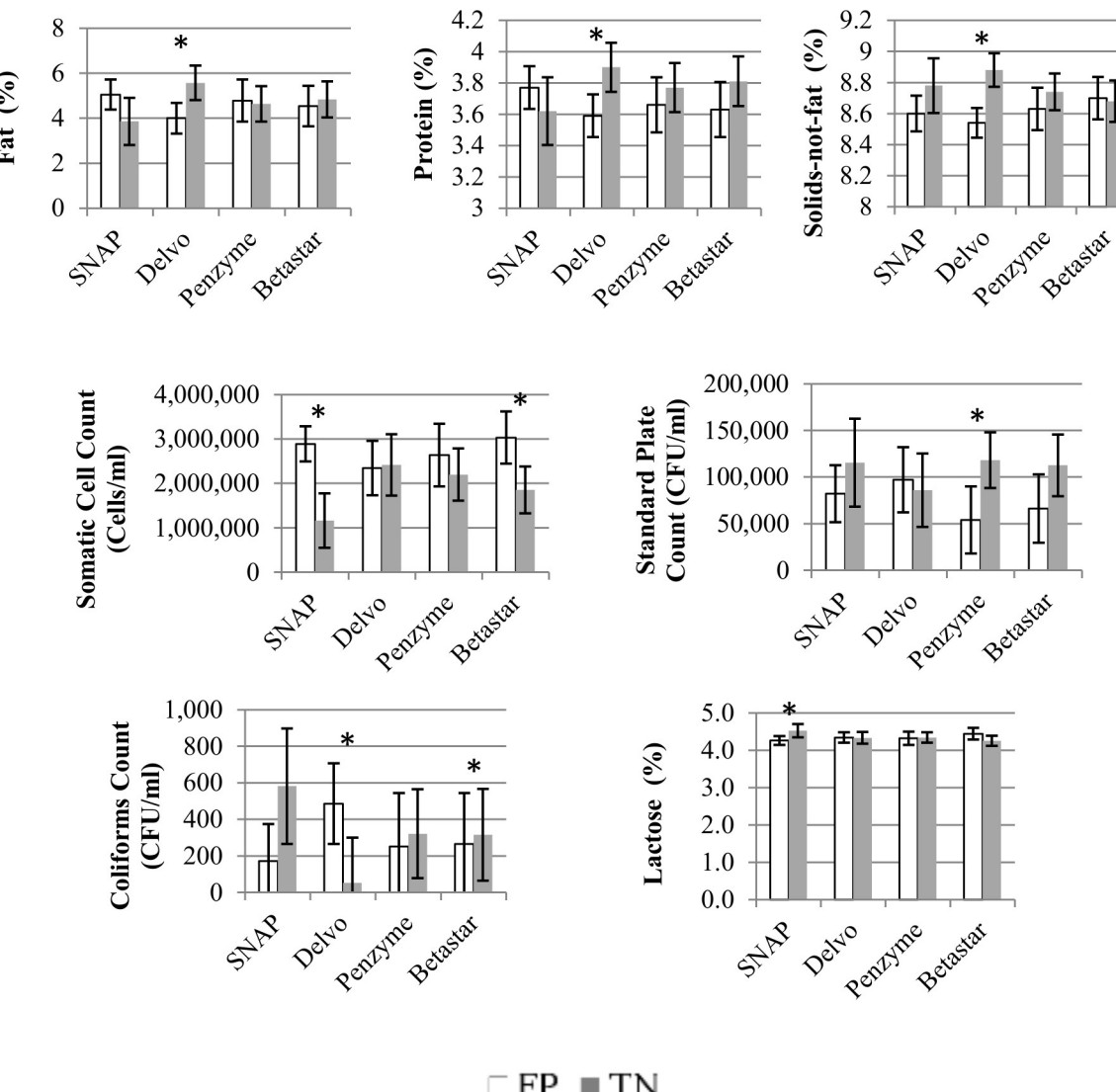

**Fig 1. Mean value for fat (%), protein (%), solids-not-fat (%), somatic cell count (CFU/ml), standard plate count (CFU /ml), coliform counts (CFU /ml), and lactose (%) for false positive (FP) and true negative (TN) test results for the four commercial tests.** Error bars correspond to 95% confidence interval. An asterisk represents test for which a significant difference was observed between FP and TN result based on Kruskal-Wallis nonparametric method.

TN results, and are summarized in **Table 3** **and S4 Table**. No false negative results were observed in the study for any of the commercial tests used.

Repeatability within individual tests for each triplicate milk sample tested revealed intra-individual test disagreement for only Penzyme and SNAP tests, with two negative test results and one positive test results for two different milk samples that were not spiked with ceftiofur, and two negative test results and one positive test result for one milk sample that was not spiked with ceftiofur, respectively.

## Discussion

Although these four commercial tests for detecting drug residues in milk are not approved for detection of drugs residues in WM samples, our findings support that all four tests detected

**Table 3. Least square means value for samples with false positive (FP) and true negative (TN) test results for variables with a significantly effect on test results.**
Only variables for which a significant difference was observed was included in the table.

| Test/variables | FP | | TN | | P-value[3] |
|---|---|---|---|---|---|
| | LSM[1] | SE[2] | LSM[1] | SE[2] | |
| **SNAP[4]** | | | | | |
| Lactose (%) | 4.26 | 0.06 | 4.52 | 0.09 | 0.037 |
| SCC (Cells/ml)* | 2,890,429 | 202,253 | 1,160,223 | 312,535 | 0.002 |
| **Delvo[5]** | | | | | |
| Fat (%) | 4 | 0.3 | 5.5 | 0.4 | 0.004 |
| Protein (%) | 3.6 | 0.07 | 3.9 | 0.08 | 0.027 |
| SNF (%)** | 8.5 | 0.05 | 8.9 | 0.05 | 0.002 |
| Coliforms (CFU/ml) | 486 | 113.4 | 52.2 | 126.7 | 0.014 |
| **Penzyme[6]** | | | | | |
| SPC (CFU/ml)*** | 53,931 | 18,484 | 118,172 | 15,310 | 0.013 |
| **Betastar[7]** | | | | | |
| SCC (Cells/ml)* | 3,032,500 | 302,032 | 1,854,000 | 270,146 | 0.027 |
| Coliforms (CFU/ml) | 265.0 | 143.3 | 315.8 | 128.2 | 0.026 |

1. Least square means for each variable found to be significantly different between FP and TN results.

2. Standard Error of the mean

3. *P*-value for Kruskal-Wallis nonparametric test evaluating a significant difference between FP and TN results for each variable.

4. Penzyme® Milk Test

5. Betastar® Plus

6. Delvo® - SP

7. SNAP® β–lactam

* Somatic cell count

** Solids-non-fat

*** Standard plate count

ceftiofur drug residues with excellent sensitivity in waste milk samples with known added drug concentrations, as reflected by a 100% sensitivity for all four tests in the study samples. When evaluating the within-individual test repeatability, most tests had high specificity, with only Penzyme and SNAP having disagreement between triplicate test results. The most common known use of WM on dairy farms is feeding it to calves, which has been shown to result in significant increases in selection for antimicrobial resistance to multiple drugs including cephalosporins, and increases in multidrug resistance in fecal *E. coli* [7]. Furthermore, recent research has also shown that feeding WM to calves impacts the development of the enteric microbiota of calves, when compared to calves fed whole milk without antimicrobial drug residues [19]. However WM is not always used as a feed source, and a recent questionnaire by the European Food Safety Authority (EFSA) Panel on Biological Hazards (BIOHAZ) showed that the disposal of WM when not fed to animals in countries in the European Union included incineration, use as an organic fertilizer, mixing in manure and spreading on land without processing, and composting [20]. These alternative approaches for disposal of waste milk could also result in selection of antimicrobial resistance in the environment, and therefore development of more effective methods for removing the undesired effects of antimicrobial residues in waste milk before disposal would be beneficial. Availability of commercial tests to detect WM samples positive for drug residues is an initial step for selecting batches of WM that may need additional treatment for degrading antimicrobial residues prior to being used as a feed source or disposed in the environment.

Delvo was the test most affected by different milk parameters. Milk samples with FP Delvo test results had lower percentages of fat percent, protein percent and SNF, and higher coliform counts when compared to samples with TN results (Table 3). Delvo test is a microbial growth inhibition assay with higher risk for FP test results linked to high SCC and the presence of the natural inhibitors (i.e., lysozyme and lactoferrin) in the samples. Furthermore, FP results have been associated with testing milk from individual animals with mastitis [10, 21–23]. Higher coliform counts observed in WM samples could be originating from fecal contamination (e.g. manure) or due to a high number of cows with coliform mastitis being milked into the WM bulk tank. Quarters with coliform bacteria have been shown to be high on lactoferrin [24]. Although in our study we did not measure lactoferrin, one explanation is that it could be responsible for the observed higher chance for FP in WM samples with significantly higher average coliform counts.

Although samples identified as FP using the Delvo test had lower protein, fat and SNF content when compared to TN samples, it must be noted that such parameters were still above the mean value expected to be found in saleable whole milk (Fig 1). For example, milk fat percent and SNF percent had mean values of 3.8% and 8.9%, respectively, in saleable bulk tank milk samples from CA in a 2017 report [25]. Furthermore, the fact that previous studies [26, 27] did not identify these factors as increasing the chance of FP test results with the Delvo test could be due to the fact that they did not evaluate these parameters within the range observed in WM samples. Furthermore, our study indicated that a higher risk for TP test results for milk protein percent, fat percent and SNF percent was observed at a specific range, which was above expected values for normal milk samples and below mean values observed in WM samples in our study. This is in contrast to expecting a linear continuous increase in FP as values for these milk parameters increase (Fig 1).

False positive SNAP test results were observed in WM samples with higher SCC (cells/ml) when compared to TN results (Table 3). The SNAP β-lactam assay identifies antimicrobial drug residues through the use of an enzyme-linked receptor-binding assay. Andrew et al. (2001) also reported that increasing somatic cells scores (SCS) were associated with an increase in FP outcomes (*P* value < 0.01) for the CITE SNAP test (IDEXX Laboratories, Westbrook, ME). In the later study, SNAP tests were evaluated using colostrum and transition milk, with mean SCC values for colostrum of 2,458,000 cell counts/ml, which is similar to the mean value observed in WM used in our study.

False positive milk samples identified using the BetaStar test had higher SCC (cells/ml) and coliform counts when compared to TN samples (S4 Table). BetaStar is a selective receptor-based beta-lactam test for penicillin-binding protein [28]. BetaStar test is based on a specific beta-lactam receptor and protein linked to gold particles, and works in a two-step phase: **1)** the preliminary incubation of a receptor with milk, will result in an interaction between the receptor and any beta-lactam antibiotics if present; **2)** the incubated medium is allowed to migrate up the dipstick, and if the receptor molecule has not been in contact with beta-lactam antibiotics, the dipstick band will capture all the receptor molecules. This will result in a visible red band being formed in the test area, interpreted as a negative test result. If no band is formed, the receptors have been blocked by beta-lactam antibiotics, interpreted as a positive test result.

In our study, our hypothesis is that high SCC could have interfered with the dipstick receptors ability to bind to receptor molecules, resulting in FP results. Specifically, the SCC interfere by reacting with the receptor molecule instead of the beta-lactam molecule. Receptor-based beta-lactam tests have been shown to be affected by high SCC (SCC > 4,000,000 cells/ml) resulting in FP results, which has been believed to be caused by natural inhibitory substances (i.e., lysozyme and lactoferrin) observed at higher concentrations in milk with high SCC [29].

Milk samples with FP test results based on the Penzyme test had lower SPC (CFU/ml) when compared to TN samples (Table 3). The Penzyme test uses an enzymatic colorimetric

technique based on an enzymatic reduction reaction within the β-lactam ring, and higher SPC could potentially interfere with this reaction [10]. Milk fat has previously been observed to increase FP test results using the SNAP test, by hindering movement of milk through the assay and causing a lack of chemical reaction [11]. Similar findings have been observed for the Penzyme test [27]. Although we did not observe that in our study, milk samples with FP test results using either the SNAP and Penzyme tests had higher mean fat percent compared to TN test results (**Fig 1**).

In addition to milk composition characteristic that could interfere with ability of the test to correctly identify positive and negative samples for drug residues, a characteristic of commercial test for drug residues in milk that could result in higher false positive results is the fact that the limit of detection for some of the β-lactam drugs tested is below the FDA tolerance or safe levels for milk (**Table 1**). Currently the FDA has established that penicillin, ceftiofur, cloxacillin, cephapirin, amoxicillin, and ampicillin are the beta-lactam drugs most commonly used to treat disease in lactating dairy cattle, and it recommends the use of a test that has been show to detect at least four of the six beta-lactams be used [15]. Tolerances and safe levels for drug residues in for milk in the U.S. can be found in the 21 code of federal regulation (CFR) 556 [30]. When establishing the limits of detection for new test, the FDA has determined that for acceptance these test shall not detect drug residues at less than 50% of the tolerance level or 25% of the target testing level for individual drugs, with the exception of penicillin G and tetracycline drug tests [9, 31]. Because of these standards for commercial drug residue test in the U.S., a false positive could occur when a drug residues is below the FDA tolerance or safe levels for milk but yet still at or above the limit of detection for the commercial test. In our study, we used the golden standard LC-MS/MS to label waste milk samples from the farm as being negative for drug residues based on FDA tolerance or safe levels for milk cut-off values, and therefore drug residues within the interval tolerance/safe and limit of detection for a test could have been due to presence of drug residue alone. An interesting finding however is that although Delvo test had the lowest detection limit to four of the six β-lactam drug screened when compared to the other three test, it did not have the lowest specificity for detection of waste milk samples without drug residues (Table 2). This supports the argument that false positive test were less likely to be caused by residues within the interval tolerance/safe and limit of detection for a test.

Divergent from the U.S. FDA approach for determining acceptable accuracy limits for commercial test screening for drug residues in milk, the European Community utilizes the Council Directive 96/23/EC as a reference for validating approaches to measure and interpret results of residues in milk [32]. Geographical differences between similar commercial tests due to local regulations should be taken into account when interpreting the findings of our study. As an example, the Council Directive 96/23/EC document includes the use of different approaches for test validation, including the use of detection capability (CCβ), which is the smallest content of the substance that may be detected, identified and/or quantified in a sample with an error probability of β, and decision limit (CCα), which is the limit at and above which it can be concluded with an error probability of α that a sample is non-compliant. As outlined in a recently published study conducted in France assessing the risk of positive reaction to widely used rapid screening test for inhibitors in milk from cows treated with a new ceftiofur formulations, CCβ and CCα are not usually clearly indicated by manufacturers of rapid drug residues tests, and because of that the lowest detected concentrations was estimated to be an appropriate threshold for the interpretation of test results [33].

## Conclusion

All four commercial tests showed 100% sensitivity for identifying samples with ceftiofur residues in inoculated WM samples. Greater variability for FP results was observed, with Penzyme

and BetaStar having the highest sensitivity for identifying TN samples. Each test was significantly affected by at least one milk quality parameter, with Delvo tests TN and FP test results affected by 4 different milk quality variables. Our findings indicate that when selecting commercial tests to detect drug residues in WM, the four commercial tests evaluated milk quality parameters must be considered if the aim is to reduce FP test results.

## Supporting information

**S1 Table. LC-MS/MS limit of quantification for drug residues in milk.**
(DOCX)

**S2 Table. Sample size output for quality parameters for waste milk samples (n = 9).** For all calculation a significant level of $\alpha = 0.05$ was used.
(DOCX)

**S3 Table. Quality parameters for waste milk samples (n = 9).**
(DOCX)

**S4 Table. Results from Kruskal-Wallis nonparametric test evaluating a significant difference between FP and TN results for each variable.** Only variables for which a significant difference was observed was included in the table.
(DOCX)

## Acknowledgments

The authors would like to acknowledge the participating dairy owners, herd managers, and veterinary practitioners. The authors also thank Kathy Glenn and Karen Tonooka from the Milk Quality Laboratory at the Veterinary Medicine Teaching and Research Center (Tulare, CA) for culture testing of milk samples.

## Author Contributions

**Conceptualization:** Marlene Belmar, Sharif Aly, Betsy M. Karle, Richard V. Pereira.

**Data curation:** Richard V. Pereira.

**Formal analysis:** Marlene Belmar, Sharif Aly, Richard V. Pereira.

**Funding acquisition:** Sharif Aly, Betsy M. Karle, Richard V. Pereira.

**Investigation:** Marlene Belmar, Sharif Aly, Betsy M. Karle, Richard V. Pereira.

**Methodology:** Marlene Belmar, Sharif Aly, Betsy M. Karle, Richard V. Pereira.

**Project administration:** Betsy M. Karle, Richard V. Pereira.

**Resources:** Marlene Belmar, Betsy M. Karle, Richard V. Pereira.

**Supervision:** Richard V. Pereira.

**Validation:** Richard V. Pereira.

**Visualization:** Richard V. Pereira.

**Writing – original draft:** Marlene Belmar, Sharif Aly, Betsy M. Karle, Richard V. Pereira.

**Writing – review & editing:** Marlene Belmar, Sharif Aly, Betsy M. Karle, Richard V. Pereira.

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
