## [Decision Letter · Decision Letter 0]

13 Aug 2019

PONE-D-19-20423

Evaluation of four commercial tests for detecting ceftiofur in waste milk bulk tank samples

PLOS ONE

Dear Dr. Pereira,

Thank you for submitting your manuscript to PLOS ONE. After careful consideration, we feel that it has merit but does not fully meet PLOS ONE’s publication criteria as it currently stands. Therefore, we invite you to submit a revised version of the manuscript that addresses the points raised during the review process.

The manuscript should be revised deeply. The main problem found in the manuscript is related to the some aspects of methodology, statistical analysis and redaction style. Please provide more reliable information on the sample collection. The manuscript should be presented according to guidelines for authors of Plos One. Thank you for your hard work.

We would appreciate receiving your revised manuscript by Sep 27 2019 11:59PM. To enhance the reproducibility of your results, we recommend that if applicable you deposit your laboratory protocols in protocols.io, where a protocol can be assigned its own identifier (DOI) such that it can be cited independently in the future. For instructions see: http://journals.plos.org/plosone/s/submission-guidelines#loc-laboratory-protocols

We look forward to receiving your revised manuscript.

Kind regards,

Arda Yildirim, Ph.D.

Academic Editor

PLOS ONE

Journal Requirements:

Additional Editor Comments:

The study is very well presented, but lacks in the quality of preparation. I agree with reviewers. The main problem found in the manuscript is related to the some aspects of methodology, statistical analysis and redaction style. Please review the referee comments and make your peer revision.

Reviewers' comments:

Reviewer's Responses to Questions

**Comments to the Author**

1. Is the manuscript technically sound, and do the data support the conclusions?

Reviewer #1: Yes

Reviewer #2: Partly

2. Has the statistical analysis been performed appropriately and rigorously? 

Reviewer #1: Yes

Reviewer #2: No

3. Have the authors made all data underlying the findings in their manuscript fully available?

Reviewer #1: Yes

Reviewer #2: Yes

4. Is the manuscript presented in an intelligible fashion and written in standard English?

Reviewer #1: Yes

Reviewer #2: Yes

5. Review Comments to the Author

Reviewer #1: Dear Editor,

Dear Authors,

First of all, I would like to thank you for the opportunity to review this manuscript on the Evaluation of four commercial tests for detecting ceftiofur in waste milk bulk tank samples. This article presents a primary scientific research in veterinary science of some interest in the current context of antimicrobial stewardship improvement.

To the best of my understanding, the objective of this study was to assess the reliability of commercial screening test kits in detecting ceftiofur residues from waste milk (WM). We agree that those tests have not been developed for this objective, but the control of bulk milk tank, supposedly free of a range of antimicrobials (inhibitors). Therefore, the most interesting feature of such tests is their negative predictive value, or their ability to tell us that the tested milk is free of contamination with a high level of confidence (in order to avoid technological and public health issue). Nevertheless, testing waste milk, when WM comes from cows at risk of having been treated with AB and because WM is massively used to feed newborn calves, is an acceptable stance. Thank you for investigating that.

This paper is well written in acceptable American English. The experiment is clearly presented, and the explanations are easy to follow. I support the publication of this article in PLOS One, provided that the authors respond to the following comments and consider revising certain points.

Major revision: Objective of this research (Abstract L21-22, Introduction L65-66)

This research is presented as an attempt to identify the ability of four commercial screening tests for detection of ceftiofur in WM. Since those tests have not been designed for this use, or the use on individual milk, the reader can ask himself about the real need of this research. So this paper is about test accuracy. The clearly exposed consequences of massive use of WM in newborn calves, and the early exposition to sub-therapeutic doses of antimicrobials suggest that it would be better to do not feed these animals with milk testing positive for inhibitors. The last sentence of the conclusion (L294-295) as well as the abstract and the introduction suggest that the objective of this research is to avoid the misclassification of healthy WM that can be discarded whereas it could be fed to calves. In case this objective exists in this research, it must be disclosed and the consequences of discarding FP WM explained. Alternatively, I suggest cutting these two sentences. Please clarify.

Major revision: Methodology, detection limits, Se, Sp

Based on the proposed methodology (true negative samples – HPLC tested, true positive samples – spiked with ceftiofur 100ppb), it is indeed possible to calculate the actual Se and Sp of the four tests. However, in the real world of milk testing, things go differently. Tests have been designed to maximize the detection of a variety of inhibitory substances, practically of all kinds for Delvotest. Many substances can be detected at concentrations far below the applicable MRL. Therefore, Se is generally good, whereas Sp does not make sense really, or it makes sense at the class level (b-lactams). You duly provided the necessary information in Table 1, the so-called detection limits (for b-lactam drugs). I agree that milk-related factors can influence these limits, and I appreciate your work for that.

However, Table 1 raises two questions.

- References are acceptable but they should be discussed, even data referring to official FDA documents. Most of the information available for these screening are provided by manufacturers. Over time, the information regarding detection limits change. There are for example several version of the SNAP test, available in different regions, at the global level. I expect a short discussion of these figures.

- The detection limit is not a totally clear concept. The detection capability (CCβ) and the decision limit (CCα) are two important performance characteristics of confirmatory methods for banned substances. For the EU we refer to Directive 2002/657/EC. Such a guidance does certainly exist for the US. I expect a short discussion of these statistical questions.

May I suggest you read a recently published paper on this matter? > doi:10.1136/vetreco-2018-000329

Minor revision: Figure 1

Figure 1 is poorly presented. It is supposed to present the core results of this study. Title and legend is required.

Typos

- L234: that (duplication)

- L404: SuPPoting information

Reviewer #2: It is unclear whether the sample size represents the population? (power analysis should be done).

The statistical tests used should be explicitly given in the method.

The normal distribution of the data can be determined by the Shapiro Wilk test. (if n<=30).especially in coliforms Count measurements, std error is higher than average, therefore non parametric methods should be used. normal distribution cannot be observed. (Mann Whitney U Test or median test can be used.)

Some typos should be corrected. (In Table S1)

The statistical method described in the abstract and the narrative in the statistical method is not similar. Specially linear regression or multiple linear regression. Which was used?

Linear regression does not evaluate a significant difference. However, it establishes the equation of the relationship between variables.

The number of samples is not sufficient for multiple linear regression. It is more appropriate that the data be at least 5 times the number of variable.

If the model is to be established, the coefficient of determination, constant and regression coefficient of the models should be given.(Table 3).

6. PLOS authors have the option to publish the peer review history of their article (what does this mean?). If published, this will include your full peer review and any attached files.

Reviewer #1: Yes: Luc DUREL

Reviewer #2: No

---

## [Author Response · Author response to Decision Letter 0]

25 Sep 2019

Reviewer #1: Dear Editor,

Dear Authors,

First of all, I would like to thank you for the opportunity to review this manuscript on the Evaluation of four commercial tests for detecting ceftiofur in waste milk bulk tank samples. This article presents a primary scientific research in veterinary science of some interest in the current context of antimicrobial stewardship improvement.

To the best of my understanding, the objective of this study was to assess the reliability of commercial screening test kits in detecting ceftiofur residues from waste milk (WM). We agree that those tests have not been developed for this objective, but the control of bulk milk tank, supposedly free of a range of antimicrobials (inhibitors). Therefore, the most interesting feature of such tests is their negative predictive value, or their ability to tell us that the tested milk is free of contamination with a high level of confidence (in order to avoid technological and public health issue). Nevertheless, testing waste milk, when WM comes from cows at risk of having been treated with AB and because WM is massively used to feed newborn calves, is an acceptable stance. Thank you for investigating that.

This paper is well written in acceptable American English. The experiment is clearly presented, and the explanations are easy to follow. I support the publication of this article in PLOS One, provided that the authors respond to the following comments and consider revising certain points.

AU: The authors thank the reviewer’s efforts during the careful and through review of the manuscript. Comments and suggestions have helped increase clarity and relevance of the manuscript.

Major revision: Objective of this research (Abstract L21-22, Introduction L65-66)

This research is presented as an attempt to identify the ability of four commercial screening tests for detection of ceftiofur in WM. Since those tests have not been designed for this use, or the use on individual milk, the reader can ask himself about the real need of this research. So this paper is about test accuracy. The clearly exposed consequences of massive use of WM in newborn calves, and the early exposition to sub-therapeutic doses of antimicrobials suggest that it would be better to do not feed these animals with milk testing positive for inhibitors. The last sentence of the conclusion (L294-295) as well as the abstract and the introduction suggest that the objective of this research is to avoid the misclassification of healthy WM that can be discarded whereas it could be fed to calves. In case this objective exists in this research, it must be disclosed and the consequences of discarding FP WM explained. Alternatively, I suggest cutting these two sentences. Please clarify.

AU: the author’s goal for this study was to evaluate the accuracy of these tests for labeling a waste milk sample as positive or not for the presence of ceftiofur residues above a tolerance limit. On a more applied and practice side, the expected use of this information would be to identify positive samples for drug residues, and utilize the appropriate intervention to reduce the undesired effect of these antimicrobial residues, be it selection of resistance when feeding to calves, or environmental selection of resistance when discarding waste milk in the fields. Currently, to my knowledge, we do not have any low cost sustainable approach that is being used for processing waste milk samples to, for example, degrade antibiotics to concentrations below those expected to be harmful. Therefore the point made by the reviewer is valid, and addressing this argument, we have added a sentence to the discussion to clarify this point, and how waste milk containing drug residues is a biological waste that, even if not fed to calves could represent a potential source of environmental selection of antimicrobial resistance. 

L 219-222- These alternative approaches for disposal of waste milk could also result in selection of antimicrobial resistance in these environments, and therefore more effective methods for removing the undesired effects of antimicrobial residues in waste milk before disposal would be beneficial.

Major revision: Methodology, detection limits, Se, Sp

Based on the proposed methodology (true negative samples – HPLC tested, true positive samples – spiked with ceftiofur 100ppb), it is indeed possible to calculate the actual Se and Sp of the four tests. However, in the real world of milk testing, things go differently. Tests have been designed to maximize the detection of a variety of inhibitory substances, practically of all kinds for Delvotest. Many substances can be detected at concentrations far below the applicable MRL. Therefore, Se is generally good, whereas Sp does not make sense really, or it makes sense at the class level (b-lactams). You duly provided the necessary information in Table 1, the so-called detection limits (for b-lactam drugs). I agree that milk-related factors can influence these limits, and I appreciate your work for that.

However, Table 1 raises two questions.

- References are acceptable but they should be discussed, even data referring to official FDA documents. Most of the information available for these screening are provided by manufacturers. Over time, the information regarding detection limits change. There are for example several version of the SNAP test, available in different regions, at the global level. I expect a short discussion of these figures.

AU: this is a very valid comment. The goal of the authors with the information in Table 1, was to address this point and have a prompt and clear resource for readers to review peculiarities of each test, especially concerning MRL’s. To address this point, the following paragraph with data from FDA documents has been added:

Ln 287-295 Currently the FDA has established that penicillin, ceftiofur, cloxacillin, cephapirin, amoxicillin, and ampicillin are the beta-lactam drugs most commonly used to treat disease in lactating dairy cattle, and it recommends the use of a test that has been show to detect at least four of the six beta-lactams be used [15]. Tolerances and safe levels for drug residues in for milk in the U.S. can be found in the 21 code of federal regulation (CFR) 556 [30]. When establishing the limits of detection for new test, the FDA has determined that for acceptance these test shall not detect drug residues at less than 50% of the tolerance level or 25% of the target testing level for individual drugs, with the exception of penicillin G and tetracycline drug tests [9, 31]. 

- The detection limit is not a totally clear concept. The detection capability (CCβ) and the decision limit (CCα) are two important performance characteristics of confirmatory methods for banned substances. For the EU we refer to Directive 2002/657/EC. Such a guidance does certainly exist for the US. I expect a short discussion of these statistical questions.

May I suggest you read a recently published paper on this matter? > doi:10.1136/vetreco-2018-000329

AU: that is a good point. Thanks for sharing the Council directive legislation and the manuscript. Both documents were inserted into the discussion of the manuscript and used to indicate that data form our study should be cautiously interpreted given regional differences for validating rapid test for detection of drug residues in milk.

Ln 306-320 Divergent from the U.S. FDA approach for determining acceptable accuracy limits for commercial test screening for drug residues in milk, the European Community utilizes the Council Directive 96/23/EC as a reference for validating approaches to measure and interpret results of residues in milk [32]. Geographical differences between similar commercial tests due to local regulations should be taken into account when interpreting the findings of our study. As an example, the Council Directive 96/23/EC document includes the use of different approaches for test validation, including the use of detection capability (CCβ), which is the smallest content of the substance that may be detected, identified and/or quantified in a sample with an error probability of β, and decision limit (CCα), which is the limit at and above which it can be concluded with an error probability of α that a sample is non-compliant. As outlined in a recently published study conducted in France assessing the risk of positive reaction to widely used rapid screening test for inhibitors in milk from cows treated with a new ceftiofur formulations, CCβ and CCα are not usually clearly indicated by manufacturers of rapid drug residues tests, and because of that the lowest detected concentrations was estimated to be an appropriate threshold for the interpretation of test results [33]. 

Minor revision: Figure 1

Figure 1 is poorly presented. It is supposed to present the core results of this study. Title and legend is required.

AU: There may be a miscommunication on how the figures are presented due to the format in which Plos One request figures be submitted. According to Plos One submission guidelines: “Figure captions are inserted immediately after the first paragraph in which the figure is cited. Figure files are uploaded separately.”

 Therefore the figures title and legend are present in the main manuscript title and follow:

Figure 1. Mean value for fat (%), protein (%), solids-not-fat (%), somatic cell count (CFU/ml), standard plate count (CFU /ml), coliform counts (CFU /ml), and lactose (%) for false positive (FP) and true negative (TN) test results for the four commercial tests. Error bars correspond to 95% confidence interval. An asterisk represents test for which a significant difference was observed between FP and TN result. 

 We hope this clarifies the comments for this figure.

Typos

- L234: that (duplication)

AU: typo corrected

- L404: SuPPoting information

AU: typo corrected

Reviewer #2: 

AU: The authors thank the reviewer’s efforts through review and suggestions and comments provided to improve the quality of the manuscript. We have carefully evaluated and addressed each point.

It is unclear whether the sample size represents the population? (power analysis should be done).

AU: information on sample size calculation was added to the manuscript.

L 86-90 Samples size calculation for detecting a difference between two means between false positive and true negative samples for all milk quality parameters was conducted using this function in JMP (SAS Institute Inc., Cary, NC). The parameters and values used for this calculation were based on our current dataset, and output of this analysis is displayed in S2 Table.

The statistical tests used should be explicitly given in the method.

AU: the statistical method section was thoroughly reviewed.

The normal distribution of the data can be determined by the Shapiro Wilk test. (if n<=30).especially in coliforms Count measurements, std error is higher than average, therefore non parametric methods should be used. normal distribution cannot be observed. (Mann Whitney U Test or median test can be used.)

AU: data was re-evaluated including the Shapiro-Wilk test as well as other assumption test. The data was re-analyzed using the Kruskal-Wallis nonparametric approach, and data is presented in the new manuscript version.

Some typos should be corrected. (In Table S1)

AU: table 1 was reviewed.

The statistical method described in the abstract and the narrative in the statistical method is not similar. Specially linear regression or multiple linear regression. Which was used?

AU: statistical method was reviewed and checked for consistency throughout the manuscript.

Linear regression does not evaluate a significant difference. However, it establishes the equation of the relationship between variables.

AU: as already mentioned, statistical method was reviewed

The number of samples is not sufficient for multiple linear regression. It is more appropriate that the data be at least 5 times the number of variable.

AU: as already mentioned, statistical method was reviewed

If the model is to be established, the coefficient of determination, constant and regression coefficient of the models should be given.(Table 3).

AU: as already mentioned, statistical method was reviewed

---

## [Decision Letter · Decision Letter 1]

24 Oct 2019

Evaluation of four commercial tests for detecting ceftiofur in waste milk bulk tank samples

PONE-D-19-20423R1

Dear Dr. Pereira,

We are pleased to inform you that your manuscript has been judged scientifically suitable for publication and will be formally accepted for publication once it complies with all outstanding technical requirements.

With kind regards,

Arda Yildirim, Ph.D.

Academic Editor

PLOS ONE

Additional Editor Comments (optional):

Thank you for taking the time to completely revise the manuscript.

Reviewers' comments:

Reviewer's Responses to Questions

**Comments to the Author**

1. If the authors have adequately addressed your comments raised in a previous round of review and you feel that this manuscript is now acceptable for publication, you may indicate that here to bypass the “Comments to the Author” section, enter your conflict of interest statement in the “Confidential to Editor” section, and submit your "Accept" recommendation.

Reviewer #1: All comments have been addressed

Reviewer #2: All comments have been addressed

2. Is the manuscript technically sound, and do the data support the conclusions?

Reviewer #1: Yes

Reviewer #2: Yes

3. Has the statistical analysis been performed appropriately and rigorously? 

Reviewer #1: Yes

Reviewer #2: Yes

4. Have the authors made all data underlying the findings in their manuscript fully available?

Reviewer #1: Yes

Reviewer #2: Yes

5. Is the manuscript presented in an intelligible fashion and written in standard English?

Reviewer #1: Yes

Reviewer #2: Yes

6. Review Comments to the Author

Reviewer #1: No other comments. This work addressed a critical point of young calf rearing and led to practical recommendations. I fully support the publication of this manuscript.

Reviewer #2: (No Response)

7. PLOS authors have the option to publish the peer review history of their article (what does this mean?). If published, this will include your full peer review and any attached files.

Reviewer #1: No

Reviewer #2: No

---

## [Editor Report · Acceptance letter]

5 Nov 2019

PONE-D-19-20423R1 

Evaluation of four commercial tests for detecting ceftiofur in waste milk bulk tank samples 

Dear Dr. Pereira:

I am pleased to inform you that your manuscript has been deemed suitable for publication in PLOS ONE. Congratulations! Your manuscript is now with our production department. 

With kind regards,

on behalf of

Dr. Arda Yildirim 

Academic Editor

PLOS ONE